# Business Model Dynamics from Interaction with Open Innovation

**Montserrat Peñarroya-Farell** * and **Francesc Miralles**

La Salle Campus Barcelona, Innova Institute, Ramon Llull University, 08022 Barcelona, Spain;
francesc.miralles@salle.url.edu
*   Correspondence: montserrat.penarroya@salle.url.edu

**Abstract:** In today's competitive environment, firms face strong challenges. We live in a volatile, uncertain, complex and ambiguous (VUCA) environment where open innovation is a strategic choice and, on top of that, the COVID-19 pandemic has emphasized most of these disrupting forces. Incumbent companies must act strategically by adapting their business model to minimize the risk and to capture the new value that emerges. This article intends to contribute to the development of the nascent stream of research that seeks to understand the evolution of Business Models through time—known as *Business Model Dynamics (BMD)*—and explores how to better align this evolution to the implementation settings of strategy. This exploratory study is built upon a meta-synthesis approach to identify, analyze, and clarify how academics have dealt with the three terms used in the Business Model Dynamics research strand: Business Model Innovation, Business Model Adaptation, and Business Model Evolution. The results of the meta-synthesis show that a disambiguation of concepts is necessary as, from an organizational learning point of view, it is required to provide a better connection between strategic value appropriation and changes on Business Models. This article contributes to the researcher and practitioner's literature on Business Model Dynamics offering a clear and rigorous definition of each term from a strategic point of view, thus preventing the conceptual incoherence and their reiterated wrong use as synonyms.

**Keywords:** business model innovation; BMI; business model adaptation; BMA; business model evolution; business model dynamics; change management; open innovation

## 1. Introduction

There is a general consensus that, for firms to remain competitive, they must continuously evolve and adapt their strategic settings to capture value from their customer needs. This is indeed more important in today's competitive environment where the VUCA conditions, open strategy choices, the pandemics and the strong disruptors that affect firm's competitiveness have an influence on strategic decisions. Sustained superior performance in these new and fast-moving environments depends crucially on the deployment and redeployment of superior strategic assets as appropriate for those environments [1]. One such asset is the firm's business model [1–6].

Business models change as managers not only innovate in business models, but also engage in more mundane adaptations in response to external changes [7]. Most researchers and managers will perceive Business Model Adaptation (BMA) as better suited to keep track of changes of incumbent firms in local markets [8], to attempt internationalization [8,9], and to simply adapt to the ecosystem evolution [10], whereas Business Model Innovation (Bmi) is perceived as directly linked to sustainable competitive advantage if successfully implemented [2–5,7,11,12]. While the message these studies convey is generally the same, they tend to employ different, often ad hoc, approaches to the definition of key concepts. There is a considerable conceptual ambiguity in strategic value appropriation in the BMD literature [13], mainly when an open innovation strategy is in place. The inconsistency

among the studies in this literature creates a generalizability problem and weakens their external validity. This conceptual incoherence diminishes the impact of the message regarding the importance of the evolution of Business Models through time, the real magnitude of the changes, the innovativeness of the process, and the attitude towards market disruptions.

Our research work is rooted in the strategic perspective that Business Model Dynamics requires different angles to understand changes in strategic value appropriation in the business model of a firm. In this sense, when incumbent firms are trying to react to either an external disruption or a change in the environment and pretend to seek the alignment of their current model with the new competitive environment, an approach based on BMA would fit better to the necessary changes for the business model [7,9,14,15]. On the contrary, if firms want to disrupt the market conditions [2,7,14,15] or the ecosystem status quo with a start-up approach [10], a specific approach based on BMI will be adequate. Finally, if an incremental and continuous innovation is required, BME will respond better to the path dynamics of the new business model. While BMA is the reaction to either an external disruption or a change in the environment, the process seeks the alignment of the Business Model with the new environment [7,16]; BMI seeks to disrupt the market conditions [3,4,10,11,17,18] or the ecosystem status quo [10].

As the scholar's knowledge on the field of BM Dynamics advances, the extant immaturity of these basic concepts adds difficulties to the progress of the field. Under the organizational learning lenses [19,20], in circumstances such as the Covid-19 pandemic where companies must face changes in their Business Models in order to appropriately capture value and survive, they deserve to confront the problem knowing the different degrees of change and adaptation that they can withstand. They can decide to adapt their Business Model, in different degrees, to the new environment or they can decide to innovate and seek out opportunities disrupting the market. In either case, they need to know, with accuracy, the different choices they have and the different outcomes they may encounter.

Furthermore, changes in business models affect strategy implementations actions. In this vein, our study tries to clarify the connections of the different BMD instances to the implementation of strategic settings for value capture. An organizational learning approach is used to shed light on these connections.

The main purpose of this exploratory research work is to deepen in a better understanding of the nature of the concept of 'Business Model Adaptation' to see how it fits in the 'Business Model Dynamics' literature. To complete this intention, this research work uses a systematic literature review [21] that explores how previous research works used both concepts, Business Model Adaptation and Business Model Innovation, in the extant literature of the strategic management area. Using a meta-synthesis research method [21] for papers published between September 2000 and December 2019, different categories on the usage of both terms appear. After an in-depth analysis, a total of 22 articles have been found eligible for this meta-synthesis. All of them use the Business Model concept in a dynamic way. Finally, all papers have been classified in seven categories and have been analyzed using an organizational learning approach.

A delineation of the different nature for BMA and BMI, in order to explain the foundations for a theoretical underpinning of BMA, and to clearly differentiate BMA from BMI flourishes as a result of these categories. This allows to connect each instance of BMD to different learning efforts in the implementation of the strategy. In this vein, this work contributes to the literature of strategy implementation by aligning the different instances of BMD to the effects on the implementation of the strategy through organizational learning.

This contribution can motivate new insights into the role that the business model concept can develop in the theoretical scene of the strategic management field. With the new categories, the underlying motivations of each concept are delineated, and for practitioners and decision-makers, new approaches can be obtained to decide on how to deal with

challenges in the competitive settings. In addition, new insights are provided to better develop this research area and to apply for sounder effects in the field.

This paper exhibits the following layout. First, we analyze the state-of-the-art on the evolution of business models through time; then, the research method applied in this study is demonstrated; next, the main key points and themes contained and extracted in the definitions of Business Model Adaptation are discussed. Finally, conclusions, limitations, and further research are outlined.

## 2. Background and State-of-the-Art

### 2.1. Competitive Challenges for Incumbent Companies

Market environment and competitive settings frequently receive disruptive shocks. Big worldwide disruptors like Amazon and Uber, new technological devices such as the Internet, Big Data and IoT, and global effects like the Covid-19 pandemic produce big changes that affect local SME firms because of the effects on the local competitive environment. Local SME firms need to react to these imported effects in their close market and competitive settings. These external effects require strategic adaptation and consequently fine-tuning the strategy implementation components that could be affected.

The firm's reaction requires to adapt their operations, stakeholder groups, alliances, positioning, value proposition, and all the rest of the components of the logics behind the strategy implementation when a change appears. Coherence of the changes requires a sound connection to the strategic settings for a correct value appropriation [22,23]. To deal with the change situation, this work proposes to use an organizational learning approach and to align the challenges of the coherence of the change to strategic learning.

In this work, we propose to use the business model perspective to understand how to deal with the effects that a firm must face when a disruption force appears and new value has to be captured. Firms must adapt their business models to new competitive settings and market drivers.

### 2.2. What Is a Business Model?

Business Model has been defined as the logics and the rationale for the implementation of the strategy. In some sense, a firm's Business Model represents how an organization creates, delivers, and captures value. Business models represent a relatively new construct and unit of analysis in the literature, receiving increasing attention over the last fifteen years [2,12,14,24–26]. Although there is no generally agreed upon definition, many contributions to the literature define it in terms of the firm's value proposition and market segments, the structure of the value chain required for delivering the value proposition, the mechanisms of value capture that the firm deploys, and how these elements are linked together in a value architecture [17,24–26]. We adopt this definition throughout this paper.

### 2.3. Business Model Dynamics

Following the works of Saebi et al. [7], the group of studies that refer to the changes occurring in existing firm's business models over time, often in response to an external trigger, can be categorized under the research stream of 'Business Model Dynamics'.

Business Model Dynamics focuses on 'how companies change and develop their business models to achieve sustained value creation through time' [7,23]. Different types of BMD have been characterized to represent different levels of strategic changes in firms due to external effects. Business Model Adaptation (BMA) is related to encompassing strategic settings to external effects with the main goal of guaranteeing economic sustainability of the firm. Business Model Innovation (BMI) refers to radically reconfiguring firm's competencies to respond to the external effects. Finally, Business Model Evolution is an incremental reconfiguration of some components of the business model to face the strategic challenges derived from the external effects. Each BMD instance represents a specific strategic value appropriation.

### 2.4. What Is Business Model Adaptation?

As far as we know, the term 'Business Model Adaptation' was used for the first time in this context by Andries and Debackere in 2007 [27]. Prior to these authors, the adaptation of a Business Model through time was often stated with the terms 'evolution', 'change', 'transformation', 'learning', 'erosion' and 'life cycles' among others, and sometimes just 'adaptation' or 'sequential adaptation' as seen in Chesbrough and Rosenbloom in 2002 [28], but never referring to 'Business Model Adaptation' as a concept and a well-established process in Business Model Dynamics.

Sometimes, the authors just mention the need to adapt business models through time in order to guarantee the economic sustainability of the organization: 'The initial business model is more of a proto-strategy, an initial hypothesis for how to deliver value to the customer, than it is a fully elaborated and defined plan of action. It results less from a carefully calculated choice from a diverse menu of well-understood alternatives, and more from a process of sequential adaptation to new information and possibilities' [28].

### 2.5. What Is Business Model Innovation?

Business Model Innovation refers to the search and development of new and sometimes disruptive modes of value proposition, creation, and capture [15,25] to disrupt market conditions [7,9], disrupt ecosystems [10], or enter a new international market [9].

As Chesbrough states 'innovation must include business models, rather than just technology and R&D' [3] innovation is not only about implementing new technology or developing new products, business models are an important asset when the intention is to disrupt a market.

The innovation in business models not only comes from inside the companies. The concept of 'open innovation', defined by Chesbrough and Bogers as a 'distributed innovation process based on purposively managed knowledge flows across organizational boundaries, using pecuniary and non-pecuniary mechanisms in line with the organization's business model' [29], can also be applied to the innovation of business models and it has had tremendous impact on research and practice [30,31].

Researchers agree that companies benefit differentially from adopting open innovation strategies [3,4,32]; however, the reasons are unclear. In this sense, Saebi and Foss specify the conditions under which business models 'are conducive to the success of open innovation strategies' [32]. To create new models and to focus on developing and successfully maintaining them, open innovation can make use of the 'developing circle of business models', defined by Yun [33].

Based on Yun and Yang [34], there are four different active business model-building processes: (a) the customer open innovation-based business model developing circle; (b) the user open innovation-based business model developing circle; (c) the social entrepreneurship-based business model developing circle; and (d) the engineer open innovation-based business model developing circle.

The impact of open innovation on the business models of public authorities has been analyzed by Finnegan and Nilsson, based on a case study of a network of municipalities in Sweden, they identify four emerging typologies of governmental transformation based on open innovation [35].

In 2006, Chesbrough introduced the concept to 'open business models' to illustrate that a closed business model can be seen as the 'starting point' and an open business model as 'the desirable end state of firm transformation' [5,36], where firms collaborate with the ecosystem by building up value and innovate their business model to make use of the emerging opportunities [36]. Saebi and Foss identify and describe four types of open business models [32].

For some years, BMI has been used as a global concept that included all aspects of Business Model Dynamics [2,7,15]. In this sense, authors like Mezger [37] affirm that Business Model Innovation can be conceptualized as a distinct dynamic capability and defined as 'the firm's capacity to sense business model opportunities, seize them through

the development of valuable and unique business models, and reconfigure the firms' competencies and resources accordingly' [37].

Five main areas of research have been identified during our literature review on Business Model Innovation:

(a)　Definitions of BMI from the lenses of different theories [2,25,38–41];
(b)　Tools to represent and to design business models as well as conceptual models [24,42];
(c)　Different archetypes and typologies of businesss models based on various criteria [43–47];
(d)　The processes and phases to implement Business Model Innovation [3,11,48];
(e)　Changing and adapting business models through time. This group of studies refers to Business Model Dynamics, the evolution and adaptation of business models. Little is known about this sub-domain and academics agree on a general feeling that a better understanding of the evolution of a Business Model through time is needed [6,7,24,28,49].

We have realized that BMI is a very consolidated concept with more than 1100 articles on the Web of Science; while BMA, with only 17 articles, requires an ad hoc study as, based on the hypotheses of this study, BMA and BMI are different concepts that refer to different phenomena, and, consequently, the differentiation of both terms can help with the understanding of strategic perspectives. Both concepts are different and can lead to confusion if they are not properly delimited. Disruptive Innovation Theory has created a significant impact on management practices and aroused plenty of rich debate within academia [15]. From its lenses, a company can have the will to disrupt the market, can be the victim of a market disruption or can be neutral towards the market, for example changing its business model to be more sustainable [50].

A third term arises from this literature review: Business Model Evolution. It is a recurrent and continuous process of adaptation of an actual Business Model to new information, internal or external, that is made available to the business [51,52]. It implies minor changes on different components of a Business Model [38] and often is part of the fine-tuning of a broader process of Business Model Innovation [53].

*2.6. Strategic Connection of Business Models Dynamics Instances*

Changes in Business Models should be related to the strategy implementation settings. Learning is necessary to adapt strategic settings and to build the logics of the strategy implementation that Business Models exhibit [19,20]. A confusion in the terms that define the planned outcome of the processes of BMA and BMI can lead to unwanted scenarios. Learning efforts have to be fitted to strategic challenges. Learning for disruption is different from learning for adaptation. Using the Argyris and Schön [19,20] approach, BMA and BME can be related to changes in firm's theory-in-use and, consequently, single loop learning efforts are necessary; however, BMI will have to be related to changes in firm's espoused theory and double loop learning efforts will be necessary.

Our research question, therefore, based on the theoretical framework of Business Model Dynamics, is "to what extent, from a strategic value appropriation perspective, is the concept of BMA different from BMI?" this is to say, to what extent the differences between these two processes justify a specific approach on strategic value appropriation? And to what extent can the similarities between both concepts lead to a misleading strategic value appropriation? This is summarized in

"To what extent should one be used and not the other to provide a sound strategic value appropriation?"

The Organizational Learning theory is going to help in clarifying these different roles for each instance related to the implementation of the strategy.

## 3. Research Methodology: Meta-Synthesis

The method chosen for this paper is a meta-synthesis research. Meta-synthesis is an integrative method for qualitative synthesis used to 'integrate, evaluate and interpret the

findings of multiple qualitative research studies' [21], in order to transform individual findings into conceptualizations and interpretations [54].

Meta-synthesis begins with a predefined research problem, a priori strategies for data collection, inclusion and exclusion criteria, data analysis, dealing with possible sources of bias, and synthesis of findings [55].

### 3.1. Why Meta-Synthesis?

There are three methods that can be used in a systematic literature review: aggregative, integrative and interpretive [21].

Integrative and aggregative methods are focused on summarizing findings of multiple qualitative research studies. Similarly, concepts employed to summarize data are assumed to be sufficiently predetermined and well specified. Aggregative methods produce effect sizes or percentages across studies (such as meta-summary) and integrative methods create taxonomies of the range of conceptual findings and provide the foundation for the development of conceptual descriptions of phenomena across studies [56].

Complementarily, interpretive methods involve considering findings across studies to generate new inductive understanding of the phenomena, events, or experiences [21]. Unlike aggregative and integrative methods, which rely on predetermined questions to guide the analysis, interpretive methods use an iterative process to explore what might be involved in similar situations and to understand how things connect and interact [55].

Given that we already have a research question 'To what extent are BMI and BMA different?' and 'When should one be used and not the other?' and also given that both concepts in focus are related to the field of Business Model Dynamics, or 'how companies change and develop their business models to achieve sustained value creation through time' [7–23], where works exist; meta-synthesis, an integrative method [21], is the most appropriate method for a systematic comparison of the terms BMI and BMA.

### 3.2. Data Collection, Inclusion and Exclusion Criteria

Meta-synthesis requires these steps for integrating findings: selection of studies, extracting findings, and abstracting findings [56,57], each explained in the following.

Articles were considered eligible for meta-synthesis based on the following criteria: published between September 2000 and December 2019; full-text article; English language; any country of the world. The searches were conducted on the main collection of the Web of Science.

As our study is concerned on the differences and similarities of the concepts 'business model innovation' and 'business model adaptation', the articles chosen for the data collection are articles that included the terms 'Business Model Adaptation', studies that includes the terms 'Business Model Innovation' and the word 'adaptation' or 'to adapt' to refer specifically to the adaptation of a business model without using the term BMA. In addition, we have also included all articles that include the terms 'Business Model evolution' jointly with the term 'Business Model Innovation' (see Table 1). We excluded those studies that are solely focused on 'business model innovation' from a non-dynamic perspective.

**Table 1.** Keywords included in the article's selection.

| Keywords | | Number of Articles |
|---|---|---|
| **'Business Model Adaptation'** | | 17 |
| 'Business Model Innovation' | Adaptation/'to adapt' | 25 |
| 'Business Model Innovation' | 'Business Model Evolution' | 5 |

After analyzing its content and the scope of application area, a total of 22 articles have been found to be eligible for this meta-synthesis. All the articles have in common the fact that, despite their differences, the processes of BMI and BMA use the Business Model concept in a dynamic, transformational manner [9,58], not as a static construct.

The excluded articles were either duplicates or articles using the terms BMI and the keyword 'adaptation' but not referring to the 'adaptation of a Business Model'.

The core contributions of this meta-synthesis are displayed in Table 2. The articles have been ordered by year of publication and the number of citations reported on Google Scholar. The table also includes whether the authors use BMI, BMA, or other related terms. In the rest of the research work, this list of papers and their authors are referred, respectively, as *core contributions* and *core authors*.

**Table 2.** Core contributions.

| Title | Author | Term Usage | Citations | Publication |
|---|---|---|---|---|
| The role of the business model in capturing value from innovation: evidence from Xerox Corporation's technology spin-off companies | Chesbrough, H. Rosenbloom, Richard S. (2002) [28] | BMI + adaptation. | 5092 | Industrial and corporate change |
| A research framework for analyzing eBusiness models | Pateli, A G Giaglis, G M (2004) [49] | BMI + 'to adapt a BM' | 484 | European journal of information systems |
| Adaptation in new technology-based ventures: insights at the company level | Andries, P; Debackere, K (2007) [27] | BMA | 122 | International Journal of Management Reviews |
| Reinventing your business model | Johnson, M W Christensen, C M (2008) [11] | BMI + adaptation | 3032 | Harvard Business Review |
| Capabilities and radical changes of the business models of new bioscience firms | Brink, Johan Holmén, Magnus (2009) [59] | BMI + adaptation | 68 | Creativity and Innovation Management |
| Business Model Adaptation as a dynamic capability: a theoretical lens for observing practitioner behavior | Dottore, AG (2009) [1] | Uses both BMI + BMA | 15 | BLED 2009 Proceedings |
| Business model innovation: Opportunities and barriers | Chesbrough, Henry (2010) [17] | BMI + adaptation | 3267 | Long Range Planning |
| Strategic development of business models: Implications of the web 2.0 for creating value on the internet | Wirtz, Bernd W. Schilke, Oliver Ullrich, Sebastian (2010) [47] | BMA | 701 | Long Range Planning |
| Business model dynamics and innovation: Re-establishing the missing linkages | Cavalcante, S., Kesting, P., Ulhøi, J. (2011) [60] | BMI + adaptation | 480 | Management Decision |
| Dynamics of Business Models–Strategizing, Critical Capabilities and Activities for Sustained Value Creation | Achtenhagen, L., Melin, L., Naldi, L. (2013) [23] | BMA | 366 | Long Range Planning |
| Business models for sustainable technologies: Exploring business model evolution in the case of electric vehicles | Bohnsack, René Pinkse, Jonatan Kolk, Ans (2014) [53] | BMI + Business Model Evolution | 425 | Research Policy |
| The changing university business model: a stakeholder perspective | Miller, K. Mcadam, M. Mcadam, R. (2014) [61] | BMI + Business Model Evolution | 164 | R and D Management |
| Toward a capability-based conceptualization of business model innovation: Insights from an explorative study | Mezger, Florian (2014) [37] | Uses both BMI + BMA | 125 | R and D Management |
| From refining sugar to growing tomatoes: Industrial ecology and business model evolution | Short, S. W. Bocken, Nancy Barlow, Claire Y Chertow, Marian R (2014) [62] | BMI + Business Model Evolution | 88 | Journal of Industrial Ecology |
| Business Model Adaptation and the Success of New Ventures | Balboni, B; Bortoluzzi, G (2015) [63] | Uses both BMI + BMA | 11 | Journal of Entrepreneurship Management and Innovation |
| Business Model Adaptation for emerging markets: a case study of a German automobile manufacturer in India | Landau, C; Karna, A; Sailer, M (2016) [9] | Uses both BMI + BMA | 27 | R&D Management |
| Design leaps: Business Model Adaptation in emerging economies | Sharma, S; Dixit, MR; Karna, A (2016) [8] | Uses both BMI + BMA | 4 | Journal of Asia Business Studies |
| What Drives Business Model Adaptation? The Impact of Opportunities, Threats and Strategic Orientation | Saebi, T; Lien, L; Foss, NJ (2017) [7] | Uses both BMI + BMA | 92 | Long Range Planning |
| Adapt and strive: How ventures under resource constraints create value through business model adaptations | Dopfer M, Fallahi S, Kirchberger M, Gassmann O. (2017) [58] | Uses both BMI + BMA | 7 | Creativity and Innovation Management |

**Table 2.** *Cont.*

| Title | Author | Term Usage | Citations | Publication |
|---|---|---|---|---|
| Valuing energy futures; a comparative analysis of value pools across UK energy system scenarios | Wegner, MS; Hall, S; Hardy, J; Workman, M (2017) [64] | Uses both BMI + BMA | 5 | Applied Energy |
| User-centered sustainable business model design: The case of energy efficiency services in the Netherlands | Tolkamp, J. Huijben, J.C.C.M. Mourik, R.M. Verbong, G.P.J. Bouwknegt, R. (2018) [65] | Uses both BMI + BMA | 15 | Journal of Cleaner Production |
| The typologies of power: Energy utility business models in an increasingly renewable sector | Bryant, ST. Straker, K Wrigley, C (2018) [66] | Uses both BMI + BMA | 5 | Journal of Cleaner Production |
| An Ecosystem-Level Process Model of Business Model Disruption: The Disruptor's Gambit | Snihur, Y; Thomas, Ll.D.W.; Burgelman, R (2018) [10] | BMI + adaptation | 3 | Journal of Management Studies |
| Business Model Adaptation in response to an exogenous shock: An empirical analysis of the Portuguese footwear industry | Corbo, L; Pirolo, L; Rodrigues, V (2018) [16] | BMA | 2 | International Journal of Engineering Business Management |
| Investigating the current business model innovation trends in the biotechnology industry | Horvath, B; Khazami, N; Ymeri, P; Fogarassy, C (2019) [67] | BMI + Business Model Evolution | 7 | Journal of Business Economics and Management |

## 4. Data Analysis

Each study was carefully read by two researchers, and findings were highlighted. As meta-synthesis is primarily 'concerned with understanding and describing key points and themes contained within a research literature on a given topic' [68], shortly after beginning to read and to analyze each document, it was possible to categorize data using in vivo and metaphorical codes. Coding was performed by two researchers and a third independent one reviewed the proposal. As organizing categories began to emerge, the data were placed into a matrix, and two dimensions and nine main key points were identified. See Table 3 and points 4.1 and 4.2.

### 4.1. Dimension 1: About the Nature of BMA

1. Is Business Model Adaptation a specific process or is it a form of BMI? Some authors believe that BMA is a form of BMI while others think that is a completely different process. In Section 4.3.1 of the synthesis, the differences in opinions are analyzed and summarized.
2. Is Business Model Adaptation innovative per se? Authors discuss the innovativeness of the BMA processes and the degree of radicalness. Section 4.3.2 is a comparison of the different opinions regarding the degree of innovation of both processes.
3. How many components must change to be considered a Business Model Adaptation? Authors discuss the scope of the change based on the different components of a business model that are affected. Section 4.3.3 is a summary of their beliefs from the point of view of how narrow or wide are the changes on the Business Model components.
4. Is BMA a continuous change or is it infrequent? The frequency of change in the process of BMA is discussed by several authors. In Section 4.3.4, the occurrence of BMA and BMI is analyzed.
5. Is BMA for start-ups or for incumbents? Authors debate to what extent the process of BMA and BMI are suitable for different types of companies. Section 4.3.5 summarizes the conveniences of BMA and BMA for start-ups and incumbents.
6. What is the attitude towards the market? Authors deliberate about the planned outcome of BMA and BMI. Section 4.3.6 illustrates the different outcomes of these two processes.

**Table 3.** Key points and themes that emerged from the coding strategy.

| Dimension | Key Points and Themes |
|---|---|
| The nature of BMA | 1.    Is Business Model Adaptation a specific process or is it a form of BMI? |
| | 2.    Is Business Model Adaptation innovative per se? |
| | 3.    How many components must change to be considered a Business Model Adaptation? |
| | 4.    Is BMA a continuous change or is it infrequent? |
| | 5.    Is BMA for start-ups or for incumbents? |
| | 6.    What is the attitude towards the market? |
| Theories to explain BMA | 1.    Business Model Adaptation through the lenses of Dynamic Capabilities theory |
| | 2.    Business Model Adaptation through the lenses of the Resource Based View |

*4.2. Dimension 2: Theories to Explain BMA*

1.    Business Model Adaptation through the lenses of the Dynamic Capabilities theory. Different authors analyze the BMA phenomena from the point of view of the Dynamic Capabilities theory. Section 4.1 summarizes their findings.
2.    Business Model Adaptation through the lenses of the Resource Based View. In Section 4.2, we summarize the findings of the authors that analyze BMA from these other lenses.

In every key point, we compare what the core contributing authors state about that theme, synthesizing findings in each point, and offering a final complete synthesis of findings at the end of the article.

*4.3. Comparing and Synthesizing (I): Stating the Nature of BMA*

This chapter compares the extent and degree of changes, of the seven different dimensions that flourished in the meta-synthesis about BMA and BMI nature.

4.3.1. Is BMA a Specific Process or Is It a Form of BMI?

A 'process' is a 'sequence of events or activities that describes how things change over time, or that represents an underlying pattern of cognitive transitions by an entity in dealing with an issue' [69]. Following this definition and given the definitions of BMI seen at the beginning of this research work, BMI is clearly a process, but is BMA a component of BMI? Or is BMA a form of BMI? We have found some discrepancies among the analyzed authors: some of the authors consider the adaptation of a business model just a component of a greater BMI process [10,11,17,28], while others consider the adaptation just a form of BMI [9,37,58,63] even an independent phenomenon [7,64].

Table 4 displays the statements of authors who believe that the adaptation of a business model is a component of a greater process of BMI.

Without using the term 'Business Model Adaptation,' authors like Chesbrough and Rosenbloom examine the need to adapt an existing business model in established (Incumbent) companies to achieve 'a sequential adaptation to new information and possibilities' [28].

**Table 4.** Business Model Adaptation as a component of BMI.

| BM Adaptation as Part of a Process of BMI | Findings | Author |
|---|---|---|
| 'Research to date is yet to satisfy the need for methods that can structure a firm's change endeavor either towards adopting a new business model or extending a current one to include new dimensions.' | Adaptation of a BM is part of the BMI process. | Pateli and Giaglis (2004) [49] |
| '( . . . ) The third is to compare that model to your existing model to see how much you'd have to change it to capture the opportunity.' | First, you create the concept of a new model, then you adapt your actual business model. | Johnson and Christensen (2008) [11] |
| This makes a 'radical' change empirically and analytically distinct from the slight alteration or adaptation of the initial business model which frequently occur within entrepreneurial ventures. | Adaptation of a business model is different from radical changes in Business Models even if they all are part of a BMI process. | Brink and Holmén (2009) [59] |
| 'Business Model Innovation is not a matter of superior foresight ex ante—rather, it requires significant trial and error and quite a bit of adaptation ex post. In fact, it is the product of extensive experimentation.' | Adaptation is part of the process of BMI. | Chesbrough (2010) [17] |

For these authors, the adaptation of a business model is just a component of a superior process of Business Model Innovation—a very common phenomenon in start-ups or companies that are searching for a way to disrupt the market. In addition, this affirmation is consistent with the extant knowledge on business models such as Teece that affirms that 'once articulated, the logic of the business model is subjected to the market test and needs to be modified and retested in face of changing environmental conditions' [70].

Figure 1 represents the concept where adaptation is part of the main BMI process.

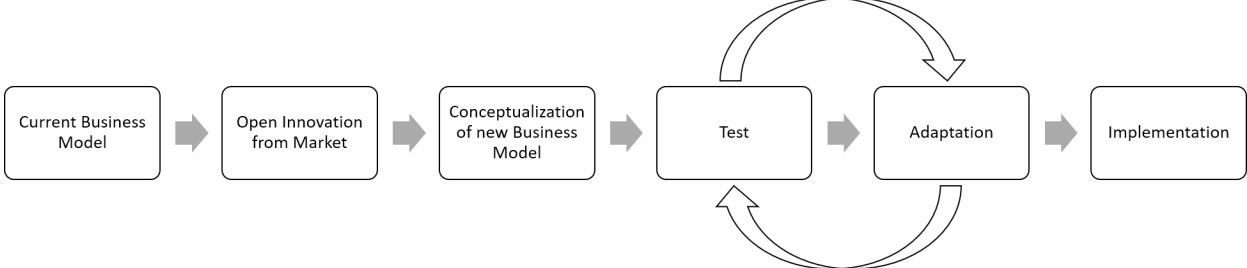

**Figure 1.** Business Model Innovation process adapted from Chesbrough and Rosenboom [28].

This same phenomenon is labeled with the term 'Business Model Evolution' (BME) in articles written after 2014 like Bohnsack et al. [53], Short et al. [71], Miller et al. [61], Balboni and Bortoluzzi [63], and Horvath et al. [67], and in a book chapter by Tina Saebi [72], which describes three different forms of business model dynamics, namely 'business model evolution, adaptation and innovation', and analyses them under the lenses of the dynamic capabilities theory.

In this article, from now on, we will use the term 'Business Model Evolution' (BME) to refer to minor adaptations of a Business Model to avoid misunderstandings and confusion with a broader phenomenon labeled as Business Model Adaptation by some of the analyzed authors.

See Table 5 to read the definitions of Business Model Evolution.

**Table 5.** Authors that define Business Model Evolution.

| Business Model Evolution | Findings | Author |
|---|---|---|
| 'Business model evolution shows a series of incremental changes that introduce service-based components ( . . . )' | BME is the creation of new BM through a series of incremental changes. | Bohnsack et al. (2014) [53] |
| '( . . . ) technology transfer office staff and government support agency representatives have led to the university business model evolving not as a process of co-creation but rather in a series of transitions ( . . . )' | BME is a series of transitions on the Business Model. | Miller et al. (2014) [61] |
| 'New ventures dynamically adapt and re-configure their business model' | BMs adapt and evolve. | Balboni and Bortoluzzi (2015) [63] |
| 'The research employs a circular evaluation method to detect which parts of the applied business structures show model evolution of an innovative and knowledge-intensive industry, biotechnology.' | Different parts of the business model show evolution. | Horvath et al. (2019) [67] |

On the other hand, some other authors think that the adaptation of a Business Model is a form of BMI as it addresses the changes of an actual Business Model to better fit a new environment [9,37,58].

See Table 6 to know the statements of authors that believe that BMA is a form of BMI.

**Table 6.** BMA is a form of BMI.

| BMA is a Form of BMI | Findings | Author |
|---|---|---|
| 'For established firms, BMI could be either the adaptation of its existing (core) business model or the development and introduction of a new business model adjacent to its core business.' | BMA is a form of BMI. | Mezger (2014) [37] |
| 'Business Model Adaptation is a form of Business Model Innovation that addresses the development of a business model to better fit a new context' | BMA is a form of BMI. | Landau et al. (2016) [9] |
| 'The process of continuous search, selection, and improvement of a Business Model based on the surrounding environment.' | The role and nature of Business Model Adaptation as a coping mechanism with resource constraints. | Dopfer et al. (2017) [58] |

Apart from the two above mentioned groups, there is a third group of authors who think that, by definition, BMA could not be BMI as the nature and the objectives of both concepts are different. For Saebi et al. [7], BMA is 'the process by which management actively aligns the firm's business model to a changing environment' [7] and Wegner et al. share the same belief [64]. We will attempt to explain the nature and the objectives of BMA, BMI, and BME in the following points.

Summary

We have realized that the longitudinal nature of any process enables to consider the adaptation of a Business Model analysis from two different viewpoints: as a phenomenon by itself where the objective is to adapt an existing business model to environmental changes; and as a component of a superior process with the objective of assessing the viability of new business initiatives. Both views are accepted among most researchers, but we realized that a new term has been coined to define the incremental adaptation of a Business Model through a series of little changes in articles written after 2014. The new denomination is Business Model Evolution.

From this meta-synthesis, we could conclude that three terms arise to describe different nuances of the processes of change in a Business Model though time:

1. **Business Model Innovation**: as the broader process with the objective to create a new Business Model.
2. **Business Model Adaptation**: as a process with the objective to adapt a current Business Model, and that can be a form of BMI if it becomes innovative.
3. **Business Model Evolution**: as a component of a wider process of transformation that seeks the change of a Business Model through small incremental changes in the current model.

### 4.3.2. Is BMA Innovative Per Se?

We have seen in the section above that authors like Saebi, Lien, and Foss consider the BMA process different from BMI [7]—not even part of it. This is because they consider that a business model can be adapted without innovation 'Business Model Adaptation and innovation differ in important ways. ( . . . ), while the kind of novelty implied by the notion of an 'innovation' might be a likely outcome of business model adaptation, it is not a necessary requirement. Business Model Adaptation can be non-innovative' [7].

In addition, authors like Sharma et al. state that BMA doesn't have to be innovative either, and affirm that it is 'quite common and a normal way of doing things' for entrepreneurs in emerging markets [8]. For them, the process consists just in adapting a model originated in a developed market, to an emerging market to better fit the environment. No innovation in the business model is needed.

For some other core authors, adapting a business model in a non-innovative way could be done, but it is a mistake 'pursuing a new business model that's not new or game-changing to your industry or market is a waste of time and money' [11].

In Table 7, we synthesize the statements about BME, BMA, and BMI regarding innovation.

**Table 7.** Innovation in BME, BMA, and BMI.

| BMA and Innovation | Findings | Author |
|---|---|---|
| 'The strategic potential of business model innovation thus lies in identifying new sources of value creation, based on innovations of the different components of a business model and/or the interactions between these components.' | BMI is based on the innovation of the different components of a Business Model. | Bohnsack et al. (2014) [53] |
| 'Entrepreneurs interested in exploring and exploiting opportunities in these markets need to overcome multiple innovation challenges to activate and sustain interest in what they have to offer.' | BMA can be innovative | Sharma et al. (2016) [8] |
| 'This article clarifies the relationship between business model innovation enabled by 3D printing technologies and the resulting innovative effect, whether radical or incremental.' | BMI is innovative (by definition) but can be either incremental or radical. | Rayna and Striukova (2016) [18] |
| 'Business Model Adaptation is a form of Business Model Innovation that addresses the development of a business model to better fit a new context' | BMA is innovative as is a form of Business Model Innovation. | Landau et al. (2016) [9] |
| 'Business Model Adaptation and innovation differ in important ways. ( . . . ), while the kind of novelty implied by the notion of an 'innovation' might be a likely outcome of business model adaptation, it is not a necessary requirement. Business Model Adaptation can be non-innovative.' | BMA can be innovative and non-innovative, while BMI is innovative. | Saebi et al. (2017) [7] |

From the perspective of its innovation, authors like Brink and Holmén state that there is a distinction between radical and incremental changes of business models. 'Radical' business model innovation arises when the business model has changed 'simultaneously within more than one aspect or dimension' [59]. They also declare that 'this makes a radical

change empirically and analytically distinct from the slight alteration or adaptation of the initial business model which frequently occur within entrepreneurial ventures' [59].

In Table 8, we analyze the degree of innovation of BME, BMA, and BMI from the perspective of our core review authors.

**Table 8.** Degree of Innovation in BME, BMA, and BMI.

| BMA and Innovation | Findings | Author |
|---|---|---|
| 'In spite of these similarities, the finding that adaptation in new ventures can imply gradual as well as radical business model changes goes against the traditional view on dynamic capabilities.' | BMA can imply gradual as well as radical changes. | Andries and Debackere (2006) [27] |
| 'The process of business model evolution involves important learning activities in which the firm develops new skills and abilities, the mind-set of innovation and adaptation, and an appetite for searching out new value creation opportunities. | The process implies incremental innovation in the firm. | Short et al. (2014) [71] |
| 'Business model evolution shows a series of incremental changes that introduce service-based components, which were initially developed by entrepreneurial firms, to the product.' | In BME the changes are incremental | Bohnsack et al. (2014) [53] |
| 'AutoLux adapted its business model in a sequential manner to step-by-step overcome the challenges of operating in an emerging market and to design a model that fits the new context' | Adaptation can be sequential to overcome step-by-step the challenges of operating in an emerging market and to design a model that fits the new context. | Landau et al. (2016) [9] |
| 'Involving the user requires facilitation of opportunities for interaction in multiple components of the business model and can lead to both incremental and radical business model innovation ex-post.' | BMI can be either incremental or radical. | Tolkamp et al. (2018) [65] |
| 'Any component of the business model can change after involving the user; however, most changes tend to be incremental changes to the value proposition and components that enable the value proposition (key activities, -resources and -partnerships).' | When adapting a BM to become user-centered, changes tend to be incremental and targeting value proposition components. | Tolkamp et al. (2018) [65] |

Summary

While Business Model Innovation is innovative by definition, Business Model Evolution and Business Model Adaptation can be innovative or non-innovative depending on the nature of its changes.

On the other hand, Business Model Evolution and Business Model Adaptation are similar from the perspective that they both normally entail organizational processes that bring about incremental adjustments to the business model, while Business Model Innovation tend to be based on radical innovation of the Business Model. Although BMA can be radical sometimes if the adaptation is innovative to the point that nothing like it has been in any other company before, and BMI can be incremental in a few cases, when for example, different phases of change are defined through the years.

### 4.3.3. How Many Components Do We Need to Change to Consider BMI and Not Just BMA or BME?

The scope of the change has been analyzed by the authors depending on the number of components that changes in the process of BMA and BMI, but authors like Balboni and Bortoluzzi, Wirtz et al., and Landau et al. agree that the number of components changed in a process of BMA is irrelevant [9,47,63], and some even state that 'at the last phase of BMA continuous adjustments of all components are required' [9].

Table 9 shows the statements of core review authors regarding the magnitude of the changes on the three processes.

**Table 9.** Changes in the Business Model components.

| Changes in Business Model Components | Findings | Author |
|---|---|---|
| 'Customer needs, market misalignments and the ability to sense new technological potential have been the major common drivers of the dynamics of these firms' BMA processes' | To succeed with the adaptation process, some components of the BM should change. | Balboni and Bortoluzzi (2015) [63] |
| 'Firms are increasingly confronted with fundamental environmental alterations, such as new competitive market structures, governmental and regulatory changes, and technological progress, which often require managers to significantly adapt one or more aspects of their business models.' | The number of aspects does not change the fact that the process is BMA. | Wirtz et al. (2010) [47] |
| 'In each phase of the Business Model Adaptation process, firms emphasize different components of the business model, before they enter into continuous adjustments of all business model components. ' | Different phases of the BMA require the adaptation of different components. At the last phase of BMA, continuous adjustments of all components are required. | Landau et al. (2016) [9] |

The core review analysis confirms that, when doing BMA, some elements of a business model should be adapted.

In our literature review, none of the authors specifies the number of components that change when the process is Business Model Evolution and is a part of a broader BMI process. However, as seen on the first point of this meta-synthesis, as this adaptation is a minor adjustment of a few components of the Business Model, and Brink and Holmen state that BME is the 'slight alteration or adaptation of the initial business model which frequently occur within entrepreneurial ventures' [59], we therefore can affirm that BME only imply changes in a few components.

#### Summary

There is no difference between the number of components that must be changed in a Business Model in a process of BMI and in a process of BMA. All components can be changed at the same time if necessary, although this will lead to a radical innovation.

As per BME, few changes are made on the components of a Business Model but with higher frequency as we will see on the next point.

### 4.3.4. What Is the Frequency of the Changes in BME, BMA, and BMI?

Business Model Adaptation as compared to BME and BMI 'takes place periodically and is likely to affect a number of business model dimensions simultaneously' [72].

Regarding BMA, Landau et al. believe that 'in each phase of the business model adaptation process, firms emphasize different components of the business model, before they enter into continuous adjustments of all business model components' [9]. Again, we can observe that by 'continuous adjustments' the authors are referring to BME. We could argue that BME is based on continuous and gradual changes of a few components of the Business Model.

Table 10 summarizes the statements of core review authors regarding the frequency of the processes of BMI, BMA and BME.

**Table 10.** Frequency of the processes of BMI, BMA, and BME.

| Changes in Business Model Components | Findings | Author |
|---|---|---|
| 'Several studies characterize business model innovation as a continuous, evolutionary process, and emphasize the role of learning in business model innovation.' | BMI is an evolutionary process. | (Landau et al., 2016) [9] |
| 'Business model adaptation involves a process of continuous search, selection, and improvement in value creation, value proposition, and value capture, based on the surrounding environment.' | BMA is a continuous process. | (Dopfer et al., 2017) [58] |

Authors like Landau et al. and Dopfer et al. argue that BMI and BMA are continuous and evolutionary processes [9,58]. In both cases, this is a vision of BMI and BMA from the lenses of Dynamic Capabilities not from the process point of view. Non-core authors agree that managers tend to avoid radical change and leave their "comfort zone", 'since such changes would require them to question their mental models and the dominant logic' [15].

Sosna et al. [73], another non-core reviewed authors, divides the business model development process into the two fundamental phases of exploration and exploitation. These two phases can be applied either to BMI and to BMA as the authors only refer to the business model development process and do not differentiate between the creation of a new business model (BMI) or de-adaptation of the existing one (BMA). We can argue that the phase of exploration of different Business Models requires a long trial-and-error-based learning, but the phase of exploitation requires stability as 'a firm cannot afford to continuously uproot, deconstruct, and innovate its extant business model' [72].

Summary

Business Model Evolution and Business Model Adaptation are similar from the perspective that they both entail organizational processes that bring about adjustments (as opposed to disruptions) to the business model. They differ, however, in the way that BME processes occur more naturally and incrementally over the lifespan of the firm's business model while BMA occurs periodically.

On the other hand, Business Model Innovation occurs infrequently as companies need a certain stability in their Business Models, but from the perspective of dynamic capabilities, both BMI and BMA, should be part of the strategic actions seeking sustained value creation in companies.

4.3.5. Is BMA for Start-Ups or Is It for Incumbents?

Core review authors analyze BMA from different perspectives regarding who is the target for such a process. Some authors consider that adaptation is a basic process for all new businesses as 'innovative business models start with an entrepreneurial idea and imagination of an offering that will serve novel value to customers. From this idea to successful implementation, new ventures experience an iterative, nonlinear, and feedback-driven process to find a match between their offering and market wants and needs' [58].

In this same line, authors like Andries and Debackere state that 'Changes to its original business model are thus needed as initially unavailable and unknown information becomes known' [52]. In addition, Balboni and Bortoluzzi are of the same mind 'In this study, we explore the connections between Business Model Adaptation and the success of new ventures' [63].

It is clear that the adaptation of their business model is a success key factor for start-ups. In addition, Andries and Debackere state that 'Especially for new technology-based firms, defining an appropriate business model from the beginning is difficult, and adaptation of the initial business model is therefore crucial for success' [52].

Authors such as Chesbrough and Rosenbloom [28] that consider adaptation a part of a BMI process even state that incumbents are not very likely to adapt their business model 'the process of adaptation appears to be either more highly motivated or more easily implemented in independent ventures than in established firms. Several of our cases suggest that the process of adaptation is triggered by the realities in the context of an independent business enterprise, which enable search processes for models far from the familiar business model of the parent company. Entrepreneurs securely employed in a large enterprise, itself with a strong culture—including its beliefs and dominant logic derived from a successful and well-established business model—may feel little incentive to search for alternatives outside that successful model' [28].

In Table 11, we address authors analyzing 'adaptation' from the point of view of start-ups.

**Table 11.** BMA is a process for Start-ups.

| Changes in Business Model Components | Findings | Author |
|---|---|---|
| 'The process of adaptation appears to be either more highly motivated or more easily implemented in independent ventures than in established firms. ' | New companies are highly motivated to change their business model. | Chesbrough and Rosenbloom (2002) [28] |
| 'Entrepreneurial firms are less constrained by path dependencies which makes them more flexible in designing more radical business models from scratch' | Entrepreneurial firms design more radical BM. | Bohnsack et al. (2014) [53] |
| 'Especially for new technology-based firms, defining an appropriate business model from the beginning is difficult, and adaptation of the initial business model is therefore crucial for success' | BME is needed for start-ups. | Andries and Debackere (2006) [27] |
| 'Companies tend to avoid major business model revisions (...) the focus on current profitable customers inhibits the exploration of emergent technologies in new commercial segments; in consequence, new business opportunities have often not been realized by incumbents, but by new ventures' | A change of BM is more likely to be done by a start-up that by an incumbent. | Cavalcante et al. (2011) [60] |
| 'A key success factor for emerging businesses of new ventures in turbulent and uncertain environments is therefore business model adaptation, characterized by rapid learning and adaptation to market changes' | Adaptation is a key success factor for new businesses. | Dopfer et al. (2017) [58] |
| '( . . . ) to reduce uncertainty about ecosystem participants' needs, entrepreneurs can adapt their business model in an effort to better meet ecosystem needs ' | Adaptation is the way entrepreneurs evolve their business model to meet ecosystem needs. | Snihur et al. (2018) [10] |
| 'In this study, we explore the connections between Business Model Adaptation and the success of new ventures'<br>'The ability to dynamically adjust the business model to changing environmental conditions and emerging market opportunities is a key capability expected to increase a start-up's likelihood of survival in the short term and to support its growth in the medium and long term' | BMA is a key factor for the success of new ventures. | Balboni and Bortoluzzi (2015) [63] |
| 'We derived a model detailing the implications of different components of disruptive innovation and unveiling how incumbents can react through BMA.' | BMA is the response of the incumbents to a disruptive innovation. | Cozzolino et al. (2018) [74] |

Dopfer et al. [58] cite Bhide that coins the term 'opportunistic adaptation' [75] to refer the phenomena where entrepreneurs adapt their business model to unexpected circumstances in an 'opportunistic' fashion as they have limited funds and have little reason to devote much effort to prior planning and research due to the high uncertainty of

their business. The author stays that 'their response derives from a spur of the moment calculation made with the intention of maximizing immediate cash-flow' [75].

Other authors consider that BMA is ideal for incumbents. This latter is the case described by Landau et al., stating that when incumbent firms enter a new market 'Firms have to innovate and adapt their business models to better fit the specific context of these international markets' [9]. In addition, it is also described by Cavalcante et al. affirming that 'the focus on current profitable customers inhibits the exploration of emergent technologies in new commercial segments' [60].

In Table 12, we address the authors that refer BMA specifically to incumbent companies.

**Table 12.** Regarding incumbents.

| Changes in Business Model Components | Findings | Author |
|---|---|---|
| 'This is an important step as there is mounting evidence of multiple threats to utility firms which require long-term business model transition and adaptation to address'. | BMA is a long-term key success factor for well-established firms. | Wegner et al. (2017) [64] |
| 'Firms have to innovate and adapt their business models to better fit the specific context of these international markets'. | BMA is a success factor when incumbents enter a new market. | Landau et al. (2016) [9] |
| 'For established firms, BMI could be either the adaptation of its existing (core) business model or the development and introduction of a new business model adjacent to its core business' | In established firms, BMA is a part of BMI. | Mezger (2014) [37] |

Other core authors believe that BMA is necessary either for star-ups or for incumbents 'Put together, our study provides specific pointers to managers and entrepreneurs looking to create opportunities in emerging markets through business model adaptation' [8].

Cavalcante et al. state that 'radical change is recognizably more difficult and stressful' [60] and remind us that, decades ago, Joseph Schumpeter emphasized the role of the entrepreneur in 'promoting new combinations which trigger economic development, and which in turn may lead to episodic instances of creative destruction' [76].

As seen in the previous point, non-core authors like Markides agree that managers of incumbent companies do not welcome radical changes and tend to avoid leaving their 'comfort zone' [15]. The academic literature suggests three exceptions to this generalization. Specifically, established firms would, on average, find it advantageous to create disruptive business-model innovations in the following circumstances:

1. When they enter a new market where entrenched competitors have first-mover advantages (e.g., Canon entering the copier market). In such a case, the new entrant must attack by breaking the rules [15,77].
2. When their current strategy or business model is clearly inappropriate and the firm is facing a crisis (e.g., Kresge introducing the discount retail concept in the 1960s and renaming itself K-Mart) [15].
3. When they are attempting to scale up a new-to-the-world product to make it attractive to the mass market [15].

Summary

Business Model Evolution implies minor adjustments in Business Models and can be applied either to start-ups or incumbents.

Business Model Adaptation is suitable for all types of companies, extant literature shows that incumbents tend to adapt their business models when changes come from an evolution of the market.

Business Model Innovation is suitable for all types of companies, but young companies are more motivated to do radical changes and to try new and disruptive ways of attacking a market to find competitive advantages, as established firms have many other alternatives

to consider, including 'investing its limited resources in adjacent markets or taking its existing business model internationally' [15].

4.3.6. The Market Makes You Change, or Are You Changing the Market?

It is well known that established and currently very successful business models cannot be understood as permanent [3,4,17]. In times of environmental change, continuous changes to, and the development of business models, are key aspects in sustained value creation and capture [23]. Otherwise, the misfit between the new context and the firm's business model would weaken the firm. Firms neglecting to adapt their business model in reaction to changes in the competitive situation or new contexts run an increased risk of failure [78].

Disruptive Innovation Theory has created a significant impact on management practices and aroused plenty of rich debate within academia [15]. As seen in the state-of-the art at the beginning of this article, from the lenses of the Disruptive Innovation Theory, a company can have the will to disrupt the market, can be the victim of a market disruption, or can be neutral towards the market, for example changing its business model to be more sustainable [50]. In our research, we have realized that this is precisely what some authors consider the main differences between BMI and BMA to be. While BMI is 'the process by which management actively innovates the business model to disrupt market conditions' [7], BMA is the reaction to a market change [7,58,72].

Furthermore, for Saebi et al., 'Business Model Adaptation and innovation differ in the important ways. ( . . . ) while Business Model Adaptation is a response to external causes, Business Model Innovation may be driven by internal as well as external factors [7] and state that 'In adapting the business model to changing external conditions, the firm aims to attain alignment with the environment' [7] while an important motivation for Business Model Innovation is to 'shape markets or industries by means of creating disruptive innovations' [7].

Core reviewed authors also state that firms are increasingly confronted with fundamental environmental alterations, such as new competitive market structures, governmental and regulatory changes, and technological progress, and therefore require managers to adapt their business models [9,47]. Other studies have linked changes in business models to unusual events in an established market. The adaptation of the business model of all industries is required; this is the case, for example, of the Portuguese footwear industry after China's entry in the WTO in 2001, analyzed by Corbo et al. [16].

However, not all adaptations are due to disrupting changes, the perception of opportunities in a new market can also engage a BMA process. Wegner et al. state that 'the combination of low barriers to entry (for incumbents) and a robust, sizeable value pool, suggests adapting utility business models to capture this revenue would be an attractive option' [64], and Achtenhagen declares that 'when companies succeed in the market with their business model and realize that there is further potential to expand, strategizing actions often lead to adaptations in the value creation logic' [23].

Landau et al. specify that 'Being able to adapt business models to different institutional settings and customer preferences are key capabilities required for firms seeking to benefit from doing business in emerging markets' [9]; therefore, not only the adaptation should be due to the result of market changes, but also to better fit a new context. In this same line, we found Sharma et al. stating that 'our main thesis of Business Model Adaptation is based on the premise that localization is necessary, and, therefore, firms need to adapt the models that are adopted from developed markets' [8].

In Table 13, we show the different motivations that drive the adaptation of a business model.

**Table 13.** BMA motivations.

| Definition | Findings | Author |
|---|---|---|
| 'Firms are increasingly confronted with fundamental environmental alterations, such as new competitive market structures, governmental and regulatory changes, and technological progress, which often require managers to significantly adapt one or more aspects of their business models.' | BMA is the reaction to environmental changes such as market, regulations, and technological progress. | Wirtz et al. (2010) [47] |
| 'Business Model Adaptation is a form of Business Model Innovation that addresses the development of a business model to better fit a new context' | Adaptation can be sequential to step-by-step overcome the challenges of operating in an emerging market and to design a model that fits the new context. | Landau et al. (2016) [9] |
| 'The process by which management actively innovates the business model to disrupt market conditions.'( . . . ) 'BMA is the reaction to a market change' | BMI is a way to disrupt a market while BMA is the reaction of a market change. | Saebi et al. (2017) [7] |
| 'Business Model Adaptation involves a process of continuous search, selection, and improvement in value creation, value proposition, and value capture, based on the surrounding environment.' | BMA is based on the changes of the surrounding environment. | Dopfer et al. (2017) [58] |
| 'While innovation, when attached to business models, is defined as the process by which firms actively innovate their business model to disrupt market conditions, the focus of this article is on how business models change in response to an external trigger. These changes have been defined as business model adaptation, that is, the process by which firms align their business model with a changing environment' | BMI aims to disrupt a market while BMA is the process by which firms align their business model to changing environments. | Corbo et al. (2018) [16] |
| 'The combination of low barriers to entry (for incumbents) and a robust, sizeable value pool, suggests adapting utility business models to capture this revenue would be an attractive option.' | The perception of opportunities in a market can drive to BMA. | Wegner et al. (2017) [64] |
| 'When companies succeed in the market with their business model and realize that there is further potential to expand, strategizing actions often lead to adaptations in the value creation logic.' | The perception of opportunities and lead to BMA. | Achtenhagen et al. (2013) [23] |
| 'Our main thesis of Business Model Adaptation is based on the premise that localization is necessary, and, therefore, firms need to adapt the models that are adopted from developed markets.' | BMA firms need to adapt the models from developed markets to better fit local environments. | Sharma et al. (2016) [8] |

Summary

When the technology push acts as an internal driver for innovation and the opportunity to disrupt the market leads to change in the business model, the phenomenon can be tagged as Business Model Innovation.

Instead, when the shift in focus goes from product solutions to customer solutions, and there are external pressures for change, that is to say, the market pulls to change the business model, the phenomenon can be tagged as Business Model Adaptation.

Regarding Business Model Evolution, the need to change could be internal or external as the need for changes arises when information that was unavailable or was unknown appears [58].

*4.4. Comparing and Synthesizing (II): Findings on Theories to Explain BMD*

This chapter summarizes the different theories that have been raised from the analysis of the different instances of Business Model Dynamics by the core review authors.

### 4.4.1. BMA and BMI as a Dynamic Capability of a Firm

Dynamic capabilities are a set of specific and identifiable processes and routines [79] that enable business enterprises to create, deploy, and protect the intangible assets that support superior long-run business performance [80].

Several non-core authors adopt the dynamic capabilities framework as a theoretical lens for observing BMI [2,12,40,81,82] and also to analyze the adaptation of business models [27,80,83,84] stating that 'if routines can be identified, then it would suggest that adaptation is indeed a dynamic capability' [27].

As shown in Table 14, in our core review, five of the papers make use of the dynamic capabilities view to explore deeper on BMA: Dottore [1], Cavalcante et al. [60], Achtenhagen et al. [23], and Balboni and Bortoluzzi [63].

**Table 14.** Dynamic capabilities view of BMA by core review authors.

| Dynamic Capabilities | Findings | Author |
|---|---|---|
| 'The dynamic capabilities framework appears to hold significant prospect for aiding the research into Business Model Adaptation and innovation.' | BMA is a determinant of sustained superior performance in fast moving and high technology markets. | Dottore (2009) [1] |
| 'If understood as part of a firm's dynamic capabilities, the adaptation of the business model to a firm's innovation activities assumes key strategic importance.' | BMA can be understood as part of a firm's dynamic capabilities. | Cavalcante et al. (2011) [60] |
| 'We employ an activity-and capability-based view on what is needed to achieve business model change.' | BMA can be analyzed from the lens of dynamic capabilities. | Achtenhagen et al. (2013) [23] |
| 'The ability to dynamically adjust the business model to changing environmental conditions and emerging market opportunities is a key capability expected to increase a start-up's likelihood of survival in the short term and to support its growth in the medium and long term.' 'The firms' dynamic capabilities have been critical in keeping them alive and kicking in three highly dynamic business environments.' | The dynamic adaptation of the business model acts as a driver of the success of the new venture. The authors analyze how three firms implemented BMA in an agile way. | Balboni and Bortoluzzi (2015) [63] |
| 'Firms create a new business model by combining, integrating and leveraging internal resources with the capabilities and resources of the ecosystem' | BMA depends on the internal resources as well as the capabilities and the resources of the ecosystem. | Sharma et al. (2016) [8] |

Summary

It seems clear that both processes BMI and BMA have in common that they have been studied through the dynamic capabilities' theoretical lens. This is even more so for the authors that consider BMA a form of BMI and therefore consider BMA a form of dynamic capability. The findings demonstrate that BMI can be conceptualized as a distinct dynamic capability. This capability can be disaggregated into a firm's capacity to sense business model opportunities, seize them through the development of valuable and unique business models, and reconfigure the firms' competencies and resources accordingly' [37].

### 4.4.2. BMA and BMI and the Resource-Based View (RBV)

Dopfer et al. [58] analyze BMA from the lenses of the resource-based view (RBV) theory. This theory, rooted in evolutionary economics, originates in the idea that a firm's sustained competitive advantage relates to the exploitation of its available resources [85].

The authors answer the question 'How do new ventures organize their business model components in order to meet their available resources?' and state that new ventures face huge challenges 'as they adapt the business model based on limited resources in order to find the product-market fit' and that 'the venture needs to go through an iterative process of adaptation to achieve complementarity between business model components and a firm's available resource base' [58].

In another of the core reviewed articles, Wegner et al. [64] also adopt a resource-based view of the firm to argue, while analyzing the evolution of the energy market in the U.K., that 'quantifying the relative size of the markets created and destroyed by energy transitions can provide useful insight into firm behavior and innovation policy' [64].

Furthermore, Landau et al. declare that 'The activity system-based view addresses business model adaptations due to institutional factors and lack of external value creation partners' [9]. In addition, Sharma et al. state that 'Firms create a new business model by combining, integrating and leveraging internal resources with the capabilities and resources of the ecosystem' [8].

Table 15 summarizes the statements of core review authors regarding BMA under the lenses of the RBV theory.

**Table 15.** BMA and the resource-based view.

| BMA the Resource-Based View | Findings | Author |
|---|---|---|
| 'The activity system-based view addresses business model adaptations due to institutional factors and lack of external value creation partners.' | This is a proper view to analyze BMA. | Landau et al. (2016) [9] |
| 'Firms create a new business model by combining, integrating and leveraging internal resources with the capabilities and resources of the ecosystem' | BMA depends on the internal resources as well as the capabilities and the resources of the ecosystem. | Sharma et al. (2016) [8] |
| 'New ventures face huge challenges 'as they adapt the business model based on limited resources in order to find the product-market fit' 'the venture needs to go through an iterative process of adaptation to achieve complementarity between business model components and a firm's available resource base' | BMA depends on the use of the limited resources of a company. | Dopfer et al. (2017) [58] |
| 'Quantifying the relative size of the markets created and destroyed by energy transitions can provide useful insight into firm behavior and innovation policy' | Resource-Based View is useful to understand a firm behavior when adapting its business model. | Wegner et al. (2017) [64] |

Summary

We observe that, as the Resource-Based View (RBV) of the firm is the proximate antecedent of the dynamic capabilities' framework [1], BMA can be analyzed from its lenses.

## 5. Discussion

BMA and BMI are complex concepts and, as we have realized in our literature review, rather poorly defined terms. The use of multiple historical definitions of Business Model Innovation (BMI) causes logical inconsistencies, self-contradictions, and conceptual ambiguity, that is to say, conceptual incoherence in the use of this term. On top of that, there are some authors that use the term 'adaptation of a Business Model' to refer to minor and recurrent changes to the Business Model, and this can be the source of confusions with the phenomenon called 'Business Model Adaptation'.

Furthermore, in addition to the effects of the conceptual incoherence of the terms to refer to the different instances of BMD, our literature review allows to propose different

roles for each instance related to the implementation of the strategy. Organizational Learning is going to help in clarifying these different roles.

In the following sections, both aspects are delimitated.

### 5.1. Conceptual Coherence of BMD Instances

The results of our meta-synthesis of extant literature have shown that three terms can be defined from different nuances of the changes that can take part in Business Model Dynamics: Business Model Evolution, Business Model Adaptation, and Business Model Innovation.

Business Model Evolution. It is a recurrent and continuous process of adaptation of an actual Business Model to new information, internal or external, that is made available to the business [3,27]. Its final objective is the maintenance and constant adaptation of a Business Model, it does not seek to disrupt the market, and it aims to preserve its relevance [63]. It implies minor changes on different components of a Business Model [61] and often is part of the fine-tuning of a broader process of Business Model Innovation [53]. All types of business can implement processes of Business Model Evolution and, in fact, from the perspective of the dynamic capabilities theory, it is advisable to continuously 'search for competitive advantages thanks to the changes on the Business Model' the constant adaptation of a Business Model should be part of the strategic actions seeking sustained value creation in companies [60,63].

Business Model Adaptation. It is a change in an actual Business Model that searches the alignment with changes in the environment [8,13,23,47,58,64]. BMA can be innovative or not, depending on the degree of novelty of the changes implemented [8,13]. If it is innovative, it can be incremental or radical [52,59]. In this process, many components of the Business Model are changed and adapted [9,47,63]. Business Model Adaptation is a process suitable for all types of companies, but incumbents are more motivated to the adaptation of their actual Business Model than to change it radically and create a new one [9,60].

Business Model Innovation. It is the process of creation of a new Business Model with the final objective to disrupt the market [7,17,58] or their ecosystem [10]. Often, the degree of innovation is radical, although it can be incremental in some cases [59]. Often, the process of Business Model Innovation implies changes in many components of the Business Model and entails the creation of new core activities and processes [9]. BMI is for all type of companies, but young companies are more motivated to implement radical changes and to try new and disruptive ways of attack a market to find competitive advantages, as established firms have many other alternatives to consider [15].

Table 16 summarizes the main characteristics of each instance of Business Model Dynamics: Business Model Evolution, Adaptation, and Innovation using the dimensions that appeared in the literature review.

### 5.2. Connection of BMD Instances to Strategy Implementation

Our research work roots on the strategic perspective that Business Model Dynamics require different perspectives to understand changes in the business model of a firm and value capture. In the previous section, our discussion allows for clarifying the delimitation of the different terms that are used for BMD instances. Each BMD instance refers to a different concept regarding the effects of the strategic settings into a firm's Business Model. In summary, each BMD instance represents a different level of participation in the implementation of the Business Model and therefore all of them are necessary and have to be used in an appropriate way.

In addition to the conceptual delimitation, each BMD instance exhibits a specific relationship regarding strategy implementation and value appropriation. At this point, our goal is to understand, from a strategic perspective, what differences can be derived among BMD instances. Taking into account that each BMD instance is a different logic or rationale for the implementation of the strategy [2,12,14,24,25,86], our work borrows the organizational learning [19,20] approach to shed some light on the strategic implementation

of each BMD instance when a change appears in the strategic settings, and new value needs to be captured.

**Table 16.** Comparing different dimensions of the processes.

| | Dimensions | Business Model Evolution | Business Model Adaptation | Business Model Innovation |
|---|---|---|---|---|
| 1 | Process or component | Component of BMI process | A process by itself but could be a form of BMI if innovative | A process by itself |
| 2 | Type of Business Model Change | Non-innovative & Innovative | Non innovative & Innovative | Innovative |
| 3 | If innovative, type of innovation | Incremental | Incremental & Radical | New BM and sometimes Radical |
| 4 | Magnitude of the changes | Few BM components are changed | Many components are changed | Many components are changed |
| 5 | Frequency of change | Continuous | Periodically | Infrequently |
| 6 | Type of companies that benefits from the process | All | All can, but incumbents could be more motivated | All can, but young companies could be more motivated |
| 7 | Attitude towards market disruption | Neutral | Victim of disruption | Seeks the disruption |

From the analysis in Section 4.4, theoretical frameworks used to study Business Model Dynamics allow for understanding that specific firm's capabilities (see Section 4.4.1) and resources (see Section 4.4.2) are in place in firms to respond to the changes on the strategic settings. The BMD actions require the question firm's mental model and leave the current "comfort zone" [15] (see Section 4.3.4), and this is necessary in the exploration and exploitation phases of the business model (see Section 4.3.5) and in reaction to technology push and market pull effects (see Section 4.3.6). All of these aspects are related to the change situation of the business model and require an action for strategic implementation.

Following the organizational learning approach, on the one hand, adjustments as in BMA and BME can be assimilated to effects on the theory-in-use [19,20] that can be implemented through single loop learning efforts. On the other hand, disruptions like in BMI can be assimilated to changes in the espoused theory and could require double loop learning efforts. Although this first analysis could explain the basic foundations between the instances of BMD and the strategy implementation, our literature review allows for postulating that this is not the case. Concretely, strategic adjustments as in BMA and BME can require leaving the comfort zone and questioning the firm's mental model and consequently requiring updating of the espoused theory. In this case, adaptation to the new strategic settings would be only possible by double loop learning efforts.

In summary, what the literature review exhibits is that a clear delimitation of the different BMD instances is necessary; however, the connection of each instance to the strategy implementation actions has to be related to learning capabilities of the firm.

## 6. Conclusions and Implications

For firms to remain competitive in today's environment—where the VUCA conditions, open innovation strategic implementation settings, the pandemics, and the strong disruptors affect firm's competitiveness—they must continuously evolve and adapt their strategic settings for a convenient value appropriation. Sustained superior performance in these new and fast-moving environments depends crucially on the deployment and redeployment of superior strategy in the firm's business model.

The aim of the paper and the purpose of this meta-synthesis is to gain knowledge and comprehension in the field of Business Model Dynamics and, firstly, to signal the evidence of a conceptual incoherence in value appropriation as a result of the use of the terms "Business Model Innovation", "Business Model Adaptation", and "Business Model Evolution" as synonyms. We consider that this is one of the causes why this phenomenon remains poorly understood, despite its importance for managers, policy makers, and academics alike. Secondly, although each BMD instance has a specific influence on adapting a business model to the new strategic settings, this delimitation is not enough to understand the success in the conversion of the new strategic challenges to sound business models. In this sense, organizational learning helps to understand the connection

of each BMD instance to strategy implementation. What our literature research exhibits is that each BMD instance can propose changes that can affect the theory-in-use or the espoused theory of the firm and to face each one of them firms should develop adequate learning capabilities.

In the book chapter 'Business Model Evolution, Adaptation or innovation?, a contingency framework on business model dynamics, environmental change, and dynamic capabilities' written in 2014, Tina Saebi describes the differences between these three terms and analyzes five different perspectives (planned outcome, scope of change, degree of radicalness, frequency of change, and degree of novelty) [72]. In our literature review, we synthesize 22 articles corroborating, in some aspects, Saebi's work, and we increase the perspectives to seven dimensions adding the type of company and the attitude towards market disruption to complete the concepts' delimitations.

Our delimitation of the three concepts BME, BMA, and BMI is wide enough to accommodate concepts like 'to pivot' [87], a metaphor widely used by practitioners meaning 'changing the business model'. Companies can pivot their business model following either the process of BMA or the process of BMI, depending on the scope of the changes, the kind of value to be captured, and the seven dimensions of our definition of BMD processes.

### 6.1. Theoretical Contributions

Our paper shows that there are nuances that escape the constraints of BMI when changing a Business Model to adapt it to fit to changes suffered by a market and to capture the new value that emerges. These two concepts differ mostly in the nature of their implementation and the final goal they seek. Therefore, it is important to use one or the other depending on the context, but not both instinctively as synonyms. This is even more interesting, in those cases where the implementation of BMI requires minor adaptations of the business model that some authors have labeled as Business Model Evolution.

We maintain there is a conceptual incoherence in the Business Model Dynamics literature with respect to the adaptation of business models because of the imprecise use of the terms 'Business Model Adaptation' and 'Business Model Innovation' and that, in order to advance in the knowledge of this field, practitioners and researchers should use the appropriate word for each concept.

In this vein, this research work provides a delimitation of the BME, BMA, and BMI concepts based on the results of a meta-synthesis of research works published from September 2000 to December 2019. Anchoring on the theories of incremental and radical innovation, disruptive innovation, dynamic capabilities, and resource-based view, the outcome of this research work can propose that BME, BMA, and BMI exhibit a behavior that has to be analyzed from a specific perspective and it makes no sense, from a conceptual endeavor, to treat each one of these Business Model Dynamics instances under the same theoretical approach. By properly describing the contents of BME, BMA, and BMI, we contribute to the research field of Business Model Dynamics.

Moreover, our work connects BMD instances to strategy implementation by using organizational learning approaches. In this vein, although business models are the rationale of strategic implementation, the different BMD instances require a specific analysis to understand the learning capabilities that a firm must develop to be able to apply the new strategic settings to the rationale behind the updated business model.

### 6.2. Limitations

As any research effort, this study is not exempt from limitations. First of all, we first acknowledge that the different characteristics of BME, BMA, and BMI will require further validation. Although we remain confident in the reliability of the differences between the three instances of Business Model Dynamics, a sounder testing would be worthwhile to consolidate the characterization of these processes. In this vein, quantitative measures of the characteristics of each process would enable a further validation of our work. Secondly, the methodological approach used in this work is based on a meta-synthesis and

a qualitative analysis of the core contributions. On the one hand, in a meta-synthesis, the scope of the research work included could have weaknesses. On the other hand, the qualitative analysis presents some limitations. In other words, future research should go beyond the methodological choices for a better consolidation of the strategic perspectives of business models. Finally, the proposal to consider BME, BMA, and BMI as different instances of Business Model Dynamics would require further validation from a strategic management perspective.

### 6.3. New Lines for Further Research

In addition to overcoming the limitations of this research work, the authors propose additional efforts to understand, first, if scenario modelling can help to understand processes of Business Model Dynamics and, second, if the different processes can be affected by contingencies' caveats.

#### 6.3.1. Scenario Modeling

Adaptation of the business model is needed when the market is disturbed by the irruption of new competitors, as seen in one of the core reviewed articles analyzing the effect of the entry of China in the Portuguese footwear industry [16], or new products as seen in the analysis of the effect of electric car in electric markets [64]. Entry of new competitors or new products are scenarios that can enlighten strategic decisions over Business Model Adaptation, but, to date, scenario modeling has not directly addressed firm strategy and behavior. Only Wegner et al., the authors of one of the reviewed articles, established a possible relationship between BMA and scenario modelling [64]. More research on both concepts would be desirable.

#### 6.3.2. BME, BMA, and BMI from the Lenses of the Contingency Theory

Different environmental conditions, such as a change in competition or a technological breakthrough, need different organizations' responses [82]. We believe that a systematic examination of what are the relevant drivers of BMA, and what kind of changes on the different components of a Business Model are required is missing to date from extant Business Model literature. In our core authors' review, we have only found BMA analyzed from the lenses of Dynamic Capabilities and from the lenses of the Resource Based view, but not from the Contingency Theory. We believe that research from this perspective would help to shed new light in this field.

**Author Contributions:** Conceptualization, methodology, validation and formal analysis of this meta-synthesis, as well as the conclusions, have been jointly developed by both authors. All authors have read and agreed to the published version of the manuscript.

**Funding:** This research received no external funding.

**Institutional Review Board Statement:** Not applicable.

**Informed Consent Statement:** Not applicable.

**Data Availability Statement:** Articles eligible for meta-synthesis where published between September 2000 and December 2019.

**Conflicts of Interest:** The authors declare no conflict of interest.

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
