# Peer review of "Business Model Dynamics from Interaction with Open Innovation"

_2199-8531, doi:10.3390/joitmc7010081_

Round 1
Reviewer 1 Report
You can find the comments and suggestions in the attachment.

Author Response
Please, see document attached.
|
R1.1 |
This manuscript is devoted to the interesting topic connected with reviews and analyses of BMI and BMA difference. However, in my opinion, there are several debatable conclusions in the article that require additional justification or correction. |
|
Authors’ comments |
Thank you very much for your opinion. We have paid specific attention to all your comments and suggestion. We hope we have solved all of them. |
|
R1.2 |
Section 4.3.6 requires revision or deletion from the article. In fact, several hundred articles on micro-level ecosystems (companies) were published in the period 2019-2020. Especially this topic is deeply researched by Chinese and Russian scientists. And the results of these studies raise doubts your conclusions on this section. |
|
Authors’ comments |
Thank you very much for this comment. After reviewing section 4.3.6, we think that this suggestion makes sense. Following your recommendation, what we did is to delete this section and make coherent the previous content to the rest of the paper. |
|
R1.3 |
(p. 18) "Business Model Adaptation is suitable for all types of companies, but incumbents are more motivated to adapt their business model as the market evolves. Business Model Innovation is suitable for all types of companies, but start-ups are more motivated and have less ties to find new ways of innovating in Business Models». I believe that these are ambiguous conclusions that assume additional research. Explain why it is startups, and for example, not technology transfer centers or innovation ecosystems, etc. |
|
Authors’ comments |
Thank you very much for this point. We adapted the manuscript for more rigorous statements. Find the new version of the summary in the text of the manuscript. |
|
R1.4 |
The few suggestions are aimed at improving the presentation and suggesting other possible discussions: The literature is dense and comprehensive, with well-chosen contributions, suitable for the objectives of the paper. The approach is interesting, with courageous and ambitious assumptions. |
|
Authors’ comments |
Thank you very much for this comment. |
|
R1.5 |
However, in the introduction section, the authors should include their contribution to the literature as well as some hints on the methodology used and the main results. |
|
Authors’ comments |
Sorry about this. We forgot to give the appropriate details in our first version of the manuscript. We have changed the introduction section. Please, see the new introduction. |
|
R1.6 |
The authors write in the Introduction section (p.3): “This contribution can motivate new insights into the role that the business model concept can develop in the theoretical scene of the strategic management field……..”. But then in the article, the interrelation of business models and strategies is not considered. Although from the practical implementation of the research results, I think it might be advisable to describe the strategies, which will be optimal for BMI and BMA. The example of this approach - multiple circular business models (CBMs), which give opportunities and challenges across a circular value chain (the Ellen Macarthur Foundation). |
|
Authors’ comments |
Thank you very much for this comment. We have introduced the interrelation of business models and strategies throughout all the article, specially in the new introduction.
|
|
R1.7 |
Section Discussion should be more developed and focused on the relevance of the results obtained correlated with similar recent studies in the field, especially in 2019-2020 years. |
|
Authors’ comments |
Thank you very much for this suggestion, we have changed point 5 and now is a discussion section focused on the relevance of the results. See 5.1 and 5.2 |
|
R1.8 |
Minor issues: Please pay attention to topic of words, punctuation marks/paragraphs/spaces, and any other typos and errors |
|
Authors’ comments |
Thank you very much for this comment. We will do our best to solve all typos and punctuation issues. |
|
R1.9 |
Final comment: In conclusion, this study focuses on a relevant topic, presents a relatively convincing structure and is written fluently. All results are clearly and appropriately presented. So, if the above suggestions could be revised, this study is recommended to be published in the journal. |
|
Authors’ comments |
Thank you very much for this comment. We hope our new version of the manuscript deserves publication in this journal. |
|
R1.10 |
Good luck with your paper, Reviewer Date of this review 14 January 2021 |
|
Authors’ comments |
Thanks a lot. |

Reviewer 2 Report
Authors must make the following corrections in the paper:
- Authors should explain better the academic contribution of the work developed. Highlighting what is innovative / original about the existing literature.
- The references cited throughout the text are not numerical. Authors must correct this.
- Table 1 is not mentioned in the text
- Authors should develop the conclusions of the work and refer in more detail to the next steps of the work
- Table 3 should be better explained
Author Response
Please see document attached.
